# Unbalanced IDO1/IDO2 Endothelial Expression and Skewed Keynurenine Pathway in the Pathogenesis of COVID-19 and Post-COVID-19 Pneumonia

**DOI:** 10.3390/biomedicines10061332

**Published:** 2022-06-06

**Authors:** Marco Chilosi, Claudio Doglioni, Claudia Ravaglia, Guido Martignoni, Gian Luca Salvagno, Giovanni Pizzolo, Vincenzo Bronte, Venerino Poletti

**Affiliations:** 1Department of Pathology, Pederzoli Hospital, 37019 Peschiera del Garda, Italy; marcochilosi@gmail.com (M.C.); guido.martignoni@univr.it (G.M.); 2Department of Pathology, San Raffaele Scientific Institute, 20132 Milan, Italy; doglioni.claudio@hsr.it; 3Department of Diseases of the Thorax, Ospedale GB Morgagni, University of Bologna, 47121 Forlì, Italy; venerino.poletti@gmail.com; 4Department of Pathology and Diagnostics, University of Verona, 37134 Verona, Italy; 5Section of Clinical Biochemistry, University of Verona, 37134 Verona, Italy; gianluca.salvagno@univr.it; 6Service of Laboratory Medicine, Pederzoli Hospital, 37019 Peschiera del Garda, Italy; 7Department of Medicine, Section of Hematology, University of Verona, 37134 Verona, Italy; giovanni.pizzolo@univr.it; 8Istituto Oncologico Veneto, IOV-IRCCS, 35100 Padova, Italy; vincenzo.bronte@univr.it; 9Department of Respiratory Diseases and Allergy, Aarhus University Hospital, 8200 Aarhus, Denmark

**Keywords:** COVID-19, IDO, post-acute COVID syndrome, PACS, SARS-CoV-2, tryptophan/kynurenine

## Abstract

Despite intense investigation, the pathogenesis of COVID-19 and the newly defined long COVID-19 syndrome are not fully understood. Increasing evidence has been provided of metabolic alterations characterizing this group of disorders, with particular relevance of an activated tryptophan/kynurenine pathway as described in this review. Recent histological studies have documented that, in COVID-19 patients, indoleamine 2,3-dioxygenase (IDO) enzymes are differentially expressed in the pulmonary blood vessels, i.e., IDO1 prevails in early/mild pneumonia and in lung tissues from patients suffering from long COVID-19, whereas IDO2 is predominant in severe/fatal cases. We hypothesize that IDO1 is necessary for a correct control of the vascular tone of pulmonary vessels, and its deficiency in COVID-19 might be related to the syndrome’s evolution toward vascular dysfunction. The complexity of this scenario is discussed in light of possible therapeutic manipulations of the tryptophan/kynurenine pathway in COVID-19 and post-acute COVID-19 syndromes.

## 1. Introduction

SARS-CoV-2 infection responsible for Coronavirus disease 2019 (COVID-19) is associated with a variability in clinical presentation and pathologic features, such as a minority of patients rapidly progressing to severe life-threatening respiratory failure requiring mechanical ventilation [1]. A proportion of COVID-19 patients suffer from post-acute sequelae, experiencing complications affecting different organs (a condition defined as “long COVID-19”, “long-haul” syndrome, or “post-acute COVID syndrome” (PACS) [2,3,4,5,6,7,8,9]. Most common symptoms often persisting for several weeks or months include systemic manifestations (fatigue, asthenia, poor concentration, wandering fever), symptoms and signs of pulmonary functional impairment (dyspnea, cough, reduced DLCO—diffusing capacity of the lungs for carbon monoxide), neuropsychiatric manifestations (sleep disturbances, cognitive dysfunction, depression, mood changes, anxiety, headache, taste, and/or smell loss), cardiac manifestations (chest pain, palpitations, tachycardia, dysrhytmias), as well as a variety of muscle-skeletal, renal, dermatological, and gastrointestinal manifestations [7,8]. The incidence of severe PACS is relatively low and most cases can resolve in less than six months, but due to the extremely large number of infected people the weight of long-COVID is a growing health concern [10,11,12,13]. Although general consensus has been reached regarding the major pathogenic mechanisms involved in the different phases and/or endotypes of acute COVID-19, the mechanisms accounting for clinical variability and the persistence of PACS symptoms are not fully understood, and different hypotheses have been formulated including autoimmune or inflammatory sequelae, persistent viral antigens, and others. The relevance of the tryptophan/kynurenine pathway has been fully recognized in different infectious diseases, but the precise role of these alterations in different clinical COVID-19 presentations and PACS is not fully understood. In this review, we describe the main data regarding these issues, and discuss the possible pathogenic role of abnormal vascular expression of enzymes that regulate this pathway (indoleamine 2,3-dioxygenases) in different COVID-19 endotypes and PACS [14,15,16,17,18].

### 1.1. Lung Involvement in Lung and PACS

The respiratory system is a major target of SARS-CoV-2 acute infection, with variable presentations from mild pneumonia to fatal Acute Respiratory Distress Syndrome (ARDS). Post-acute lung sequelae have been described in survivors of severe COVID-19 pneumonia, as well as in people recovering from either hospitalized or non-hospitalized mild COVID-19, with a risk that increases across the clinical severity [19,20,21,22]. Data from analysis of laboratory tests, High Resolution CT scans, and lung tissue obtained from patients with lung sequelae might provide useful information regarding both acute and post-acute COVID-19 pneumonias, revealing pathogenic mechanisms that occur and develop independently of the viral presence by nature. A variety of pulmonary clinical manifestations and radiological features of interstitial disease have been observed in PACS, characterized by frequent (up to 50%) clinico-radiological features of organizing pneumonia [19,23,24,25]. The most frequent functional abnormality in PACS is lung related decreases in diffusion capacity, followed by restrictive defects [20,26,27,28,29]. A possible explanation has been hypothesized, based on abnormalities of vascular volumes occurring in PACS lung [30]. There is general agreement on considering lung vascular abnormalities and vascular dysfunction as central factors in the pathogenesis of severe COVID-19 pneumonia, although consensus has not been reached on the mechanisms of their development. In particular, the role of direct endothelial infection by SARS-CoV2 is controversial, and uncertainty remains regarding the mechanisms of ventilation/perfusion (V/Q) mismatch leading to abnormal perfusion and hypoxemia in different COVID-19 endotypes [31,32,33,34]. The pathogenic links relating vascular abnormalities occurring in COVID-19 and PACS are currently unknown. To address these issues, we recently used lung histology and immunohistochemistry to investigate a series of transbronchial lung cryobiopsies from patients with persistent symptoms and computed tomography suggestive of residual lung disease after recovery from Sars-CoV-2 infection [35]. A variety of relevant changes were observed, ranging from minimal abnormalities to fibrosing interstitial disease. An intriguing finding in PACS cases of this series was the occurrence of morphological and immunophenotypical changes in the pulmonary vascular bed, similar to those observed in acute early/mild cCOVID-19 pneumonias (vascular enlargement and abnormal endothelial expression of IDO1, PD-L1 and STAT3) [35,36]. The persistence of these peculiar findings after virus clearance (as demonstrated by molecular analysis in lung tissue, BAL, and nasal swab in all PACS cases) strongly suggests that this phenotype occurs independently from active infection, and it is likely involved in the pathogenesis of both COVID-19 and post-COVID-19 sequelae. In our opinion, the abnormal expression of the enzyme indoleamine 2,3-dioxygenase (IDO1) by endothelial cells in acute and post-acute COVID-19 pneumonias deserves particular attention; this review addresses the topic. 

### 1.2. IDO and the Tryptophan/Kynurenine Pathway

L-tryptophan (Trp) is a semi-essential amino acid utilized in protein synthesis and as a precursor of metabolites involved in a variety of important physiological mechanisms including pregnancy, neuronal function, and immune tolerance [37]. Trp consumption for protein synthesis is minimal, since >90% Trp is degraded through different pathways including the “serotonin” pathway (representing less than 10%) and the Tryptophan/Kynurenine Pathway (TKP) that largely predominates [38,39]. The TKP, regulated by a variety of enzymes expressed in different organs and conditions, determes the balanced concentration of Trp metabolites that can vary at the local and systemic levels [40]. The liver, where tryptophan-2,3-dioxygenase (TDO) is constitutive, is the predominant site for Trp degradation under physiological conditions, whereas extra-hepatic Trp degradation gains priority in inflammatory and immune activation, utilizing the catalytic activity of indoleamine-2,3-dioxigenases (IDO) [41]. The alternative serotonin pathway of Trp metabolic degradation is active in specialized sites (e.g., the brain and the pineal gland) where tryptophan hydroxylase can catalyze Trp hydroxylation, producing serotonin and melatonin [39]. The variety of Trp metabolites produced by TKP activation include kynurenine, anthranilic acid, kynurenic acid, 3-hydroxykynurenine, xanthurenic acid, 3-hydroxyanthranilic acid, quinolinic acid, picolinic acid, and finally NAD+ (a fundamental coenzyme for physiological processes such as DNA repair, cell growth, and energy metabolism). Trp metabolites have diverse biological properties, and their concentration may exert relevant roles in physiological and pathological mechanisms. The level of Trp metabolites in different body compartments is rigorously regulated, and the evaluation of blood concentration of kynurenine and the kynurenine/tryptophan ratio are considered as reliable markers of overall TKP activity at the systemic and local levels [42]. When the correct enzyme balance is altered (as in infection or cancer), the production of metabolites is modified, which affects several physiological functions [39,43].

The TKP-regulated availability of amino acids can limit the proliferation of some pathogenic microorganisms, and this competition has a protective role against infections in different species [44,45,46,47]. In mammals, this simple competitive strategy has evolved, providing novel functions in the regulation of immunity and other relevant physiological mechanisms [47,48]. 

### 1.3. Indoleamine 2,3-Dioxygenase (IDO1 and IDO2)

Indoleamine 2,3-dioxygenase enzyme activity, firstly identified as a Trp degrading enzyme in rabbit intestine [49], is the rate-limiting step of Trp degradation in extra-hepatic sites, where IDO is the main regulator of TKP activity, Trp consumption, and the production of Trp metabolites. Two closely related tryptophan catabolizing enzymes have been discovered, IDO1 and IDO2 [50,51,52]. IDO1 is highly induced by various inflammatory stimuli in different cell types and tissues, and its complex immunomodulatory functions are involved in physiologic and pathologic situations including maternal tolerance, inflammatory restraint in infection, tumor immune escape, neurodegenerative disorders, and autoimmune disorders [39]. IDO1 expression in normal tissues is negligible, but inflammatory stimuli can trigger its expression, mainly mediated by IFN [39,53,54]. Further positive and negative signals for IDO1 expression are also provided by Trp, nitric oxide (NO), H_2_O_2_, IL-6, and other cytokines [40,46,55,56,57,58].

IDO1 has relevant role in immune regulation, suppressing effector T-cell functions and favoring the development of regulatory T cells by different mechanisms including Trp depletion at the local site of inflammation and the production of immunosuppressive Trp metabolites (kynurenine, kynurenic acid, xanthurenic acid). Further immune-regulatory signals are provided by the increase in uncharged Trp-tRNAs (determined by local Trp depletion), a molecular mechanism that is able to activate the amino-acid sensitive GCN2 stress-kinase signaling, with eventual cell cycle arrest and/or anergy in T cells [59,60]. Essential amino acid deficiency can also interfere with the mTOR functional activity in dendritic cells, inducing the conversion of naïve T cells into Tregs [61,62]. 

Trp metabolites kynurenine and kynurenic acid are agonist ligands of the aryl hydrocarbon receptor (AHR), inducing T-cell apoptosis and favoring Treg development [63,64]. AHR is a multifunctional helix-loop-helix “biosensor” activated by a variety of naturally occurring and synthetic molecules, such as exogenous toxic compounds (e.g., polycyclic aromatic hydrocarbons), kynurenine, and other endogenous molecules [65]. The AHR response to exogenous or endogenous ligands can be divergent, either acting as a sensor to “danger signals” (favoring a proinflammatory Th17 response) or containing the inflammation by favoring suppressive Treg responses [65,66,67,68,69]. A further level of complexity is provided by the presence in the non-catalytic small domain of IDO1 proteins of immunoreceptor tyrosine-based inhibitory motifs (ITIMs), whose phosphorylation is dependent on the interaction with regulatory molecules; these interactions can lead to the suppression of cytokine signaling 3 (SOCS3) and the activation of the Src kinase [70,71]. In the presence of proinflammatory IL-6, proteosomal degradation of the enzyme occurs, thus interrupting tolerance [72,73]. IDO1 can in fact mediate its own expression and activity on the basis of microenvironmental molecular milieu (IL-6 and TGF-b) [65,72,74,75,76]. 

IDO2. IDO1 first appeared in placental animals through the duplication of the more ancestral paralog IDO2 gene (both located adjacent on chromosome 8) [50,77]; it is necessary to maintaining tolerance and providing protection to the fetus from T-lymphocytes [51,78,79]. 

Although IDO1 is genetically homologous to IDO2, the two enzymes have distinct expression patterns and roles: IDO2 exerting robust pro-inflammatory activity in autoimmune functions acting on B cells, whereas IDO1 is able to mediate the suppression of T cells, in opposition to the functions associated with autoimmunity [80,81,82,83,84]. Opposite functions are also exerted by IDO2 enzymes in experimental liver injury mediated by kynurenine and AHR signaling [85]. This intriguing functional difference may be related to the scarce catalytic activity of IDO2 in metabolizing TrpA compared to IDO1, and IDO2 may in fact represent a pseudoenzyme [52,86,87]. Pseudoenzymes (proteins that despite their evolutionarily similitude to active enzymes do not exert significant catalytic activity) have peculiar functions as regulators of relevant biological mechanisms; these functions include the control of substrates’ availability for their analog enzymes [88,89,90,91,92,93,94,95]. Another relevant difference is the lack of complete/functional ITIM sequences and signaling functions in IDO2, thus restraining microenvironmental regulation for its expression that mainly depends on aryl hydrocarbon receptor (AHR) rather than on interferons [73,96]. 

### 1.4. Kynurenine and Trp Metabolites as Biomarkers

The quantitative evaluation of kynurenine and kynurenine/tryptophan ratio is widely considered a reliable marker of TKP activation, and these values are perturbed in a variety of pathologies including COVID-19. Both kynurenine and K/T ratio are promising as diagnostic and prognostic biomarkers. Nevertheless, due to the large variety of cell types involved in enzyme regulation of Trp metabolism in different organs, the pathogenic significance of these biomarkers in different human diseases is not easy to decipher. This is particularly true when therapeutic manipulation of TKP abnormalities are hypothesized in different clinical contexts [92,93,94,95,96]. In general, TKP-related circulating biomarkers in the population can vary according to several parameters (age, gender, body mass index, physical activity, smoking, diabetes), and their cumulative effects likely determine the final physiologic or pathologic significance. In fact, TKP alterations of TKP-related biomarkers are associated with the risk of cancer and cardiovascular disease mortality independently of the cause [97,98,99,100]. 

Abnormal variations of kynurenine levels and K/T ratio in blood have been described in different pulmonary pathologies including infectious diseases (seasonal influenza, community-acquired pneumonia, pneumocystis infection, tuberculosis), lung transplantation and organ rejection, chronic inflammatory lung diseases, pulmonary hypertension, autoimmune diseases, and lung cancer [101,102,103,104,105,106,107,108,109,110,111]. 

### 1.5. Kynurenine and Trp Metabolites in COVID-19 and PACS

Over the past few months, a large number of studies has focused on the metabolic abnormalities occurring in COVID-19 patients, and significant changes in amino acid, lipid, and energy metabolism have been described. The methodological approaches in these studies were different, including sophisticated metabolomic analyses, but all studies documented the abnormalities in the tryptophan metabolism, as further supported by the increase in kynurenine and K/T ratio in the peripheral blood of SARS-CoV-2 positive patients [112,113,114,115,116,117,118,119,120]. These metabolic alterations had significant value as prognostic biomarkers to predict an increased risk of mortality, corresponding to different COVID-19 endotypes with increasing clinical severity [121,122,123,124,125]. Interestingly, derangement from normal values matched the occurrence of other inflammatory biomarkers (IL-6, CRP) [118,126,127]. 

Kynurenine signaling through the AHR may induce cell senescence andcontribute to aging-related pathologies of the musculoskeletal system, which can also complicate COVID-19 and PACS [128,129,130]. Modulation of the KTP with eventual systemic release of neuroprotective Trp metabolites occurs in skeletal muscles by physical exercise, and can partly explain the beneficial effects of physical exercise in different conditions, such as PACS [131,132,133]. 

In COVID-19 and PACS, several neurological complications can be observed, from isolated anosmia and/or dysgeusia to severe neuropsychiatric conditions, and the activation of TKP has been proposed as a mechanistic explanation and a promising therapeutic target [134,135,136,137,138,139,140]. In fact, the persistence of an abnormal K/T ratio and tryptophan decrease as well as vascular abnormalities occur in a number of patients and might be considered a feature of PACS [141,142,143]. The activation of TKP has been generally considered a possible cause of the neurological complications occurring in SARS-CoV-2 infection [118,134]. An abnormal KTP activation can significantly interfere with neurological physiology by decreasing the availability of the essential amino acid Trp for conversion to 5-HT and melatonin (molecules necessary for regulation of sleep, mood, and appetite), as well as by producing unbalanced proportions of neurotoxic (quinolinic acid, 3-hydroxykynurenine) versus neuroprotective and anti-depressive (kynurenic acid, picolinic acid, and the essential cofactor NAD+) Trp metabolites [94,144,145,146,147]. Within the brain, quinolinic acid concentrations are normally lower compared to blood, but IDO1-expressing dendritic cells, microglia, and macrophages raise the levels of the neurotoxic quinolinic acid during inflammation [148,149,150]. Accordingly, abnormal increases in the concentration of kynurenine and kynurenin/tryptophan ratio are observed in neurodegenerative and neuropsychiatric disorders [151,152,153,154]. 

### 1.6. IDO Expression in Cells and Tissues

The expression pattern and tissue distribution of the different enzymes involved in the TKP (TDO, IDO1, IDO2) has been evaluated in different species utilizing a variety of methodological strategies, and a detailed picture in human tissues is still partial [155]. Human IDO1, as in other mammals, has a restricted distribution, likely related to its distinctive functions [79]. IDO1 is mainly expressed in placenta (where the enzyme is considered to exert a relevant role in the maternal–fetal tolerance process) [78,156], in lymphoid tissues, both intestine and lung [157,158]. In lymph nodes and the thymus, dendritic cells with antigen presenting functions are the main cell type expressing IDO1 in immunohistochemical investigations [159]. 

Consistent IDO1 immunohistochemical expression has been described in the lung, mainly confined to blood vessels [159]. The endothelial expression of IDO1 in other tissues is absent in normal conditions, but when up-regulated by IFNg the resulting vascular deprivation of Trp may provide antibacterial activity [160]. In our experience, the endothelial IDO1 expression in a normal lung is weak and/or restricted to scattered blood vessels [35,36]. A significantly different IDO1 expression pattern is observed in COVID-19 early/mild pneumonia and PACS patients, where most parenchymal blood vessels, both capillaries and venules, consistently show endothelial immunostaining (Figure 1) [35,36]. According to available data, IDO1 and IDO2 have overlapping but distinct functions and expression patterns. IDO1 expression in immune cells is variable and dependent on cytokines’ availability in the microenvironment, whereas IDO2 is constitutive in circulating myeloid DCs and plasmacytoid dendritic cells [161]. According to the few available studies, IDO2 expression in a normal lung is negligible [162]. 

### 1.7. IDO1 Regulation of the Vascular Tone: Lessons from Placental Pathology and Pulmonary Hypertension

In the human placenta, IDO1 is constitutively expressed in chorionic vascular endothelium, with the highest levels found in the microvasculature. The endothelial expression increases in distribution from first trimester to term, paralleling the high increase in the kynurenine-to-tryptophan ratio occurring in chorionic villous tissue [163,164]. The IFNg secreted by Natural killer cells at the maternal–fetal interface can likely significantly contribute to local constitutive induction of IDO1 [165]. In addition to the well-established role in maintaining feto–maternal immune-tolerance and antimicrobial functions, IDO1 has been demonstrated to exert a relevant role in the regulation of vascular tone and placental perfusion, thus providing a regular blood flux to the growing fetus [166]. Accordingly, in experimental IDO1 deficiency, a number of pregnancy disorders can develop such as impairments in intrauterine growth restriction (IUGR) and pre-eclampsia [166,167,168,169,170]. Kynurenine is an endogenous relaxing factor for blood vessels [92,171], and the TKP has been proposed as a therapeutic target in pre-eclampsia and other hypertensive disorders [172,173]. Further complexity has been recently evidenced, since IDO1 can regulate vascular tone in inflammation by producing singlet molecular oxygen and the Trp metabolite cis-WOOH (cis-hydroperoxide (2S,3aR,8aR)-3a-hydroperoxy-1,2,3,3a,8,8a-hexahydropyrrolo[2,3-b]indole-2-carboxylic acid) [174,175]. Endothelial IDO1 is likely necessary for exerting vascular relaxation, and critical levels of endothelial and/or perivascular concentration of vasoactive Trp metabolites may be necessary for effective control of the vascular tone [176]. Some Trp metabolites can easily diffuse and enter into cell cytoplasm, but some specific functions (i.e. the vascular tone control) likely depend on the actual local availability, and the “average” concentration can have different significances in different microenvironments. The activity of endothelial IDO1/kynurenine axis on vascular tone is likely more effective in organs characterized by peculiar circulatory systems such as the placenta and lung, both characterized by reduced blood pressure and both constitutively expressing IDO1 [159,177]. 

Increased kynurenine concentration in the vessel microenvironment can selectively operate in pulmonary hypertension by contrasting apoptosis in endothelial cells, but favoring apoptosis in smooth muscle cell [178]. In pulmonary hypertension, the observed increase of kynurenine serum levels, concomitant with Trp decrease, may serve as a negative feedback mechanism contrasting vascular pressure increase, and its negative prognostic significance as a biomarker can be treated as evidence of an insufficient protective effect on vascular dysfunction [179,180]. On the other hand, several studies have demonstrated a central role of IDO1 activity and kynurenine in inducing systemic vascular relaxation and hypotension in experimental and human septic shock [181,182,183,184,185]. An endothelial protective role of IDO1 has also been demonstrated in experimental ischemia-reperfusion, atherosclerosis, and acute lung allograft injury, thus suggesting a role for the TKP activation in mitigating vascular dysfunction and conditions exhibiting both excessive inflammation and a compromised balance between vasoconstrictor and vasodilator tone [186,187,188,189,190]. 

### 1.8. Vascular Dysfunction, COVID-19 and the TKP

Vascular dysfunction is associated with a variety of pathological conditions (cardiovascular disease, diabetes, obesity, older age, chronic lung disease, infections, etc.), and is a distinct feature of severe COVID-19 [191,192,193,194]. In fact, SARS-CoV-2 infection can induce vascular inflammation, disruption of the endothelial homeostasis, edema, and life-threatening coagulation abnormalities in severe cases, all features defining endothelial/vascular dysfunction [195,196,197,198,199]. The pathophysiology of endothelial dysfunction in COVID-19 is generally considered a consequence of the body’s uncontrolled inflammatory response, but the precise mechanisms accounting for its development have not been completely defined [31,193,200,201,202]. Direct infection has been considered a possible cause, but evidence of SARS-CoV-2 endothelial infection has only been rarely reported in pulmonary and extra-pulmonary sites [203,204,205], and experimental endotheliopathy can be triggered by plasma from severely ill patients [206]. Independent of viral infection, the interaction of SARS-CoV-2 spike proteins with different cell types may represent a plausible cause of endothelial damage, senescence, and impairment of endothelium-dependent vasodilation, thus representing a pathogenic trigger in COVID-19 and PACS [18,102,207,208,209,210,211,212,213]. 

### 1.9. IDO1 and IDO2 in COVID-19 Pneumonia: From Vasoplegia to Vascular Dysfunction?

In situ analyses have allowed a precise evaluation of IDO1 expression at the cellular level in acute early/mild COVID-19 pneumonias, revealing an intense and diffuse immunoreactivity in blood vessels, both capillaries and venules, at variance with what has been observed in control normal-lung samples and in a variety of other pulmonary pathologies; yet these results are similar to what has been observed in human placenta (Figure 1) [36]. In the same samples, SARS-CoV2 viral RNA was demonstrated in alveolar epithelial cells, concomitantly with IL-6 signals and STAT3 over-expression. Furthermore, a diffuse enlargement of interstitial vessels was noted, suggesting a pathogenic role of IDO1 in inducing COVID-19 V/Q mismatch and silent/happy-hypoxia [31,32]. These findings were associated with mild pneumonia, since all cases of that series did not need invasive ventilation [36]. Comparable endothelial IDO1 expression is not observed in pulmonary autoptic cases of severe COVID-19 pneumonia (Figure 1g,h; unpublished data). Interestingly, in fatal cases that enabled investigations through necropsy, strong and diffuse pulmonary expression of IDO2 was demonstrated, with no significant evidence of IDO1 [162]. Two harmful Trp-metabolites (3-hydroxy-anthranilic acid and quinolinic acid) were co-localized with IDO2 in the same samples, suggesting that most of the enzymatic activity was due to IDO2 [162]. A possible explanation of this unprecedented hyper-expression of IDO2 was proposed, centered on a peculiar positive feedback loop generated by the interaction of kynurenine and AHR, which favored the expression of IDO2 [91]. Different endogenous and exogenous AHR ligands can induce opposite effects in immunity: dendritic cells accumulation of kynurenine results in tolerogenic signals, whereas in pollution enhanced Th17 differentiation is observed via the AHR [214]. Due to this double-face behavior, AHR may be considered a determining force in lung pathology [215,216,217]. In COVID-19, AHR-binding environmental pollutants may amplify inflammation and contribute to disease severity [218,219]. The observed imbalance of IDO1 and IDO2 expression in COVID-19 may be ascribed to different mechanisms. The two enzymes differently respond to metabolic inhibitors and cytokines (in particular, IFNg, as suggested by their differential expression in malaria and influenza) [51,220,221,222,223]. Degradation of IDO1 may be ascribed to IDO2-mediated hyper-activation of the STAT3/IL-6 pathway that trigger the enzyme proteolysis [224,225]. Extensive evidence has been provided on the pathogenic role of STAT3/IL-6 signaling in COVID-19, and therapies blocking this pathway are utilized to avoid disease evolution [226,227]. A further possibility is provided by viral activation of AHR in an IDO-independent manner [228]. Finally, the progressive decrease in IDO1 may further up-regulate IDO2 expression [229]. These different mechanisms might be responsible for a vicious circle in which the abnormal accumulation of extra-vascular kynurenine may cause the AHR and IDO2 expression to switch off endothelial IDO1. In this scenario, the harmful products of TKP activity might exert their pathogenic effects in a microenvironment deprived of the protective role of IDO1 in endothelial cells. These findings open relevant issues regarding the molecular mechanisms occurring in COVID-19 pneumonia, and the role of predisposing conditions in development of vascular dysfunction in severe cases. 

We propose a pathogenic model in which the differential expression of IDO1 and IDO2 in COVID-19 pneumonia has pathogenic and prognostic relevance. In this model two different conditions are hypothesized: (1) Early/mild COVID-19 pneumonia where the activation of inflammatory signals (i.e IFNg) induce vascular IDO1, TKP activation, and production of potential harmful Trp metabolites. The protective role of IDO1 is preserved, thereby avoiding further vascular damage. (2) Severe COVID-19 pneumonia where predisposing conditions (age, diabetes, obesity, etc.) provide a background activation of AHR, thereby leading to IDO1/IDO2 imbalance and the switch from protective vasodilatation to vascular dysfunction, as previously described. Interestingly, acute increase of kynurenine is able to induce loss of vascular-tone control and endothelial dysfunction [230]. Impairment of vascular-tone regulation and bioavailability of vasodilators (including IDO1 metabolites and NO) is considered central in the development of endothelial dysfunction and diffuse alveolar damage in different conditions, such as COVID-19 [231,232,233,234,235]. The early administration of inhaled NO has been considered a possible therapeutic approach for reducing pulmonary vascular resistance and enhancing the ventilation/perfusion matching [236]. IDO1 and NO have interconnected functions and reciprocal regulation in endothelial cells. They are both induced by IFNγ and participate in a complex feedback mechanism, where interaction with NO triggers IDO1 degradation through the proteasome pathway [237]. In addition, some Trp metabolites can regulate NO production, and IDO1 has a nitrite reductase activity likely involved in observed local production of NO under anaerobic conditions [238,239]. On the other hand, evidence of IDO2 acting as a vasodilator is lacking, and it is not expressed in placental and pulmonary vessels as IDO1. A further support to the pathogenic relevance of IDO1/IDO2 imbalance may be provided by their opposite roles in shaping the immune tolerance and susceptibility to autoimmune conditions [240]. In several experimental conditions IDO1 is protective, whereas IDO2 is pro-inflammatory and is able to mediate autoreactive responses [80,82,83,241,242,243,244]. Autoimmune complications are common in COVID-19, and the possible role of IDO1/IDO2 imbalance in their development warrants further investigation [245,246].

In summary, different morphological and immunophenotypical vascular patterns can be defined in COVID-19, where the IDO1+ and IDO1-/IDO2+ phenotypes may correspond, in our view, to the previously defined biphasic presentation of COVID-19 pneumonia, with an early type-L pattern, characterized by vascular enlargement, preserved compliance, hypoxemia, and an out-of-proportion hypocapnia; and a more severe and potentially fatal type-H pattern, characterized by reduced vascular relaxation and vascular dysfunction [31,247,248,249,250]. 

### 1.10. IDO1 Endothelial Expression in Post-COVID-19 

Diffuse endothelial IDO1 expression and vascular enlargement were also observed in post-COVID pneumonia cases, independent from the severity of pulmonary pathologic pattern [35]. This finding opens possible scenarios for this still poorly defined condition, scenarios such as the neurological complications and the persistent respiratory impairment. In our view, persistently inflammatory stimuli in these patients may maintain endothelial IDO1 expression and TKP activity, despite the complete viral clearing. The persistent TKP activation may be able to maintain elevated kynurenine blood levels, thus explaining the neurological and immunological dysfunction, as well as the “encephalomyelitis/chronic fatigue syndrome” -like symptoms observed in PACS [251,252,253]. The endothelial IDO1-enzyme activity in PACS pneumonia could partly explain respiratory symptoms due to persistent Q/V mismatch as observed in mild COVID-19 pneumonia [32].

A schematic description of this hypothetical pathogenic scenario is described in Figure 2.

The pathogenic role of IDO1/IDO2 imbalance in COVID-19 and PACS should of course be considered within a wider and more complex scenario where different mechanisms occur, including other metabolic abnormalities involved in immune regulation [254,255,256]. Further studies regarding the expression pattern and functional activity of IDOs and TKP in different clinical presentations of COVID-19 and PACS are warranted.

### 1.11. Therapeutic Considerations

The complex involvement of the TKP activation in the diverse presentations of COVID-19 and PACS poses relevant issues when the possible manipulation of this pathway is considered as a therapeutic option [117]. The complex roles (protective versus harmful) of endothelial TKP activation in different physiological systems and tissues need to be carefully considered in therapeutic planning in different clinical contexts. In COVID-19, the TKP activation may be considered harmful in increasing the systemic concentration of kynurenines and toxic Trp-metabolites, potentially interfering with immune and neurological functions, as previously described. Nevertheless, the endothelial IDO1 expression likely exerts significant protection against vascular dysfunction in the lung, and under this context the inhibition of IDO1 does not appear to be safe, which means that alternative interventions are needed to correct metabolic abnormalities. The correct balance of Trp-metabolites is relevant for maintaining healthful functions, and specific approaches should be selected for treating different COVID-19 clinical presentations. This can be particularly true in PACS, where mild “supportive” therapeutic interventions may ameliorate symptoms and shorten the disease course. 

Trp supplementation, either dietary or non-nutritional, has been experimentally investigated and proposed to ameliorate neurological disturbances and social behavior in humans by increasing 5-HT production, but its clinical effect is still controversial and prone to genetic variations [257,258,259,260,261,262,263,264,265]. In addition, potential side effects of excessive Trp intake should be evaluated in different clinical settings [266]. Deranged activation of the TKP is common in COVID-19, and Trp supplementation may potentially increase the systemic concentration of harmful Trp metabolites [267].Melatonin supplementation. Divergence from the serotonin pathway of Trp metabolism induces a defective production of melatonin in COVID-19 patients, and this decrease has prognostic significance [268]. This deficiency may be either indirectly related to the over-activation of the TKP (eventually leading to lower available 5-HT), to the well-documented anti-oxidant and anti-inflammatory roles of melatonin, or both [134,269,270,271,272]. A possible role in protecting pulmonary endothelial cells can be also hypothesized, as observed in pre-eclampsia [273]. Several authors have proposed melatonin as a useful therapeutic tool in contrasting neurological complications and super-infections for both COVID-19 and PACS patients [274,275,276,277,278], and a significant vantage in mortality and recovery rate has been observed in severe cases [279,280,281]. Validation of the therapeutic role of melatonin and its metabolites in COVID-19 and PACS is needed, and clinical trials and the use of reliable animal models are warranted [282,283,284,285,286,287].IDO inhibitors. The relevant pathogenic and prognostic roles of the different Trp-metabolites in COVID-19 suggest the possible use of specific inhibitors to modulate the enzymes that regulate the TKP [117,288,289]. IDOs and TDO are involved in tumor immunosurveillance and the potential use of inhibitors of these enzymes to restore antitumor immunity is a matter of intense clinical investigation [96,290,291]. Different specific inhibitors are in fact available, and clinical trials are ongoing on a variety of human neoplastic and non-neoplastic diseases [292,293,294,295]. If the pathogenic shift from early/mild to severe COVID-19 pneumonias is in part determined by the modulation of IDO1 and IDO2 expression in the pulmonary microenvironment as here hypothesized, the availability and selective use of specific IDO1 or IDO2 inhibitors might be crucial. A range of selective and potent TDO, IDO1, and IDO2 inhibitors are currently under investigation in cancer research and information from this field may be translationally utilized for new personalized therapies for patients suffering from COVID-19 and PACS [52,86,296,297,298,299]. The inhibition of AHR is feasible, and this approach might be safer in severe cases [300].

## 2. Conclusions

Although the pathogenesis of COVID-19 and PACS is complex and only partially understood, several lines of evidence have been provided on the relevant role of immune mechanisms in triggering the abnormal cascade of cytokine production in severe cases. The regulation of these mechanisms is mediated by a variety of factors, such as genetic background and occurrence of predisposing metabolic abnormalities. The Tryptophan/kynurenine pathway is central in the regulation of immunre responses and vascular tone and may represent a key factor in the development of vascular dysfunction in severe COVID-19 pneumonia. The possible pharmacological manipulation of this pathway in SARS-CoV-2 infectious diseases should be based on the precise understanding of the different pathogenetic and clinical contests, avoiding potentially harmful consequences in the vascular compartment. 

## Figures and Tables

**Figure 1 biomedicines-10-01332-f001:**
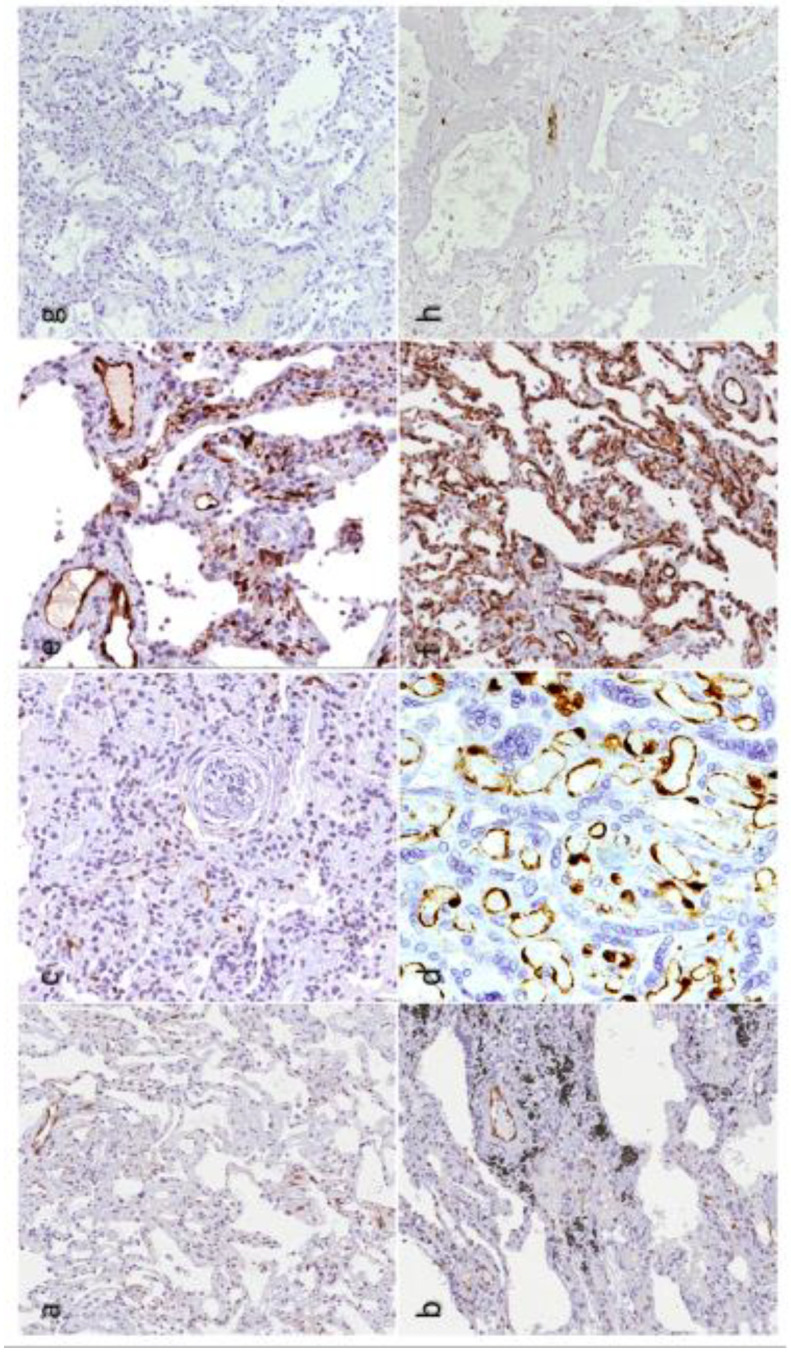
IDO1 endothelial expression is minimal in lung controls, where only scattered vessels are positive (**a**,**b**): normal lung; (**c**): organizing pneumonia. In human placenta, diffuse and strong endothelial IDO1 expression is observed in all vessels (**d**). Diffuse and strong endothelial IDO1 expression in a case of COVID-19 early/mild pneumonia (**e**). Diffuse and strong endothelial IDO1 expression in a case of post-COVID-19 pneumonia (**f**). Minimal/absent IDO1 endothelial expres-sion in two autoptic cases of severe COVID-19 (**g**,**h**). All cases were retrieved from the archive of Pathology Dept. of the San Raffaele Hospital, Milan, Italy and were immunostained with an-ti-IDO1 rabbit monoclonal antibody (dil.1:100, clone D5J4E, cod 86630, CellSignal, Danvers, MA, USA) with the Benchmark Ultra Instrument (Ventana-Roche). Original magnification in all images: 200×. Matched rabbit isotype control (cod. 3900 CellSignal, Danvers, MA, USA) on the same sections was always negative.

**Figure 2 biomedicines-10-01332-f002:**
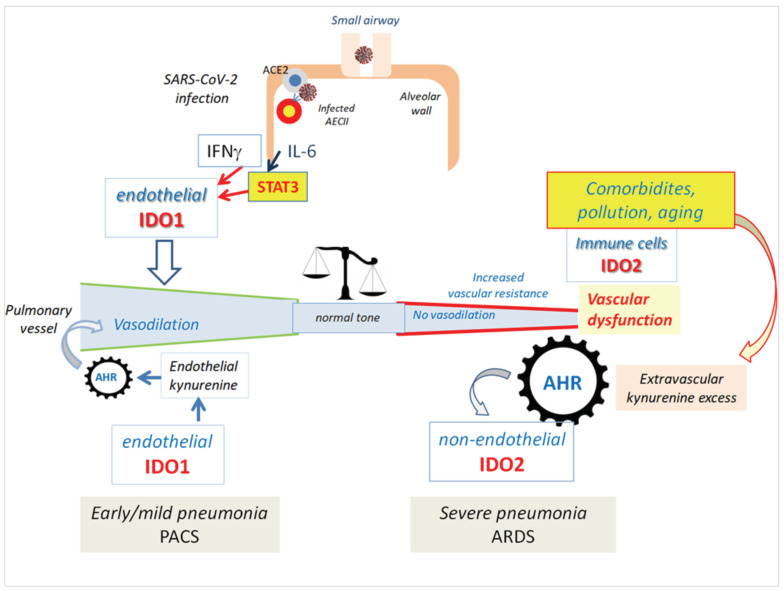
Hypothetical mechanisms involved in COVID-19 pneumonia, as discussed in this review. After SARS-CoV-2 infection leading to early/mild pneumonia, inflammatory stimuli trigger endothelial IDO1 expression, kynurenine accumulation, and vascular relaxation. This mechanism may persist in post-COVID-19. In severe cases, a loss of vascular IDO1 expression is observed, likely resulting in impairement of vascular-tone control and induction of vascular dysfunction. The occurrence of antecedent abnormal AHR activation (related to old age, comorbidities, and/or pollution) may concur in altering the kynurenine levels and the switch from IDO1 to IDO2.

## Data Availability

Not applicable.

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
