# Peer review of "Unbalanced IDO1/IDO2 Endothelial Expression and Skewed Keynurenine Pathway in the Pathogenesis of COVID-19 and Post-COVID-19 Pneumonia"

_biomedicines, 2022, doi:10.3390/biomedicines10061332_

Round 1
Reviewer 1 Report
The manuscript entitled "Unbalanced IDO1/IDO2 endothelial expression and skewed 2 keynurine pathway in the pathogenesis of Covid-19 and post- 3 Covid-19 pneumonia" draws attention to a significant problem.
However, it would be necessary to better underline the main purpose of the review, in abstract and in the introduction section. Hence, especially the introduction section should be rewritten.
When the literature check was not systematically done it may lead to missing of some already published articles, like for example,: Murakami, Yuki et al. “Remarkable role of indoleamine 2,3-dioxygenase and tryptophan metabolites in infectious diseases: potential role in macrophage-mediated inflammatory diseases.” Mediators of inflammation vol. 2013 (2013): 391984. doi:10.1155/2013/391984 or Sawada L, Vallinoto ACR, Brasil-Costa I. Regulation of the Immune Checkpoint Indoleamine 2,3-Dioxygenase Expression by Epstein-Barr Virus. Biomolecules. 2021 Nov 30;11(12):1792. doi: 10.3390/biom11121792. PMID: 34944437; PMCID: PMC8699098, that may also be worth to discuss in the context of viral infection.
There is lack of references in the Figure 1 caption. Please add the source.
I would suggest adding subheadings to the "Therapeutic considerations" section
Moreover, a seperate conclusion section would be strongly needed.
Author Response
Dear Reviewers and Editors,
Thank you so much for the opportunity to revise and resubmit our manuscript (biomedicines-1738458) entitled “Unbalanced IDO1/IDO2 endothelial expression and skewed 2 keynurine pathway in the pathogenesis of Covid-19 and post- 3 Covid-19 pneumonia” for consideration in Biomedicines.
We appreciate the reviewers for their insightful comments and we have corrected our original manuscript according to the reviewers’ comments and our point-by-point responses to the reviewer’s comments are included.
Thank you for your consideration.
We look forward to a favorable reply
On behalf of all authors
Sincerely
Claudia Ravaglia
Responds to the reviewer’s comments:
Reviewer: 1
The manuscript entitled "Unbalanced IDO1/IDO2 endothelial expression and skewed 2 keynurine pathway in the pathogenesis of Covid-19 and post- 3 Covid-19 pneumonia" draws attention to a significant problem. However, it would be necessary to better underline the main purpose of the review, in abstract and in the introduction section. Hence, especially the introduction section should be rewritten.
The introduction was changed according to the Reviewer’s suggestion.
When the literature check was not systematically done it may lead to missing of some already published articles, like for example: Murakami, Yuki et al. “Remarkable role of indoleamine 2,3-dioxygenase and tryptophan metabolites in infectious diseases: potential role in macrophage-mediated inflammatory diseases.” Mediators of inflammation vol. 2013 (2013): 391984. doi:10.1155/2013/391984 or Sawada L, Vallinoto ACR, Brasil-Costa I. Regulation of the Immune Checkpoint Indoleamine 2,3-Dioxygenase Expression by Epstein-Barr Virus. Biomolecules. 2021 Nov 30;11(12):1792. doi: 10.3390/biom11121792. PMID: 34944437; PMCID: PMC8699098, that may also be worth to discuss in the context of viral infection.
These relevant references have been added as suggested by the Reviewer.
There is lack of references in the Figure 1 caption. Please add the source.
The Figure 1 caption has been changed according to the Reviewer’s suggestion
I would suggest adding subheadings to the "Therapeutic considerations" section
Subheadings have been added according to the Reviewer’s suggestion
Moreover, a separate conclusion section would be strongly needed.
A separate conclusion has been added according to the Reviewer’s suggestion
Reviewer 2 Report
Authors provided interesting opinions on the possible role of the IDO1/IDO2 and keynurine pathway in the pathogenesis of Covid-19 and post- 3 Covid-19 pneumonia. Since that is hypothesis, not established fact, I recommend changing of the title in the way implying that (e.g. "Possible role.....")
It would be beneficial if authors provide more detailed information on mRNA and protein expression of molecules in question, as well as their distribution regarding healthy tissues.
It is not clear whether microphotographs constituting Figure 1 are authors' own, obtained in their laboratory. If so, as a part of figure caption authors mut provide details on primary and secondary antibodies (origin, isotype, dilution, manufacturer) and technique used. Also, negative (isotype) controls for each tissue/case should be shown.
Some refinement of wording is needed. I.e. :
Line 244: "....deprivation of Trp mayt (=may) provide...."
Line 246: "...IDO1 expression in normal lung is faint..." , in my opinion, instead of faint expression better choice might be weak expression.
Author Response
Dear Reviewers and Editors,
Thank you so much for the opportunity to revise and resubmit our manuscript (biomedicines-1738458) entitled “Unbalanced IDO1/IDO2 endothelial expression and skewed 2 keynurine pathway in the pathogenesis of Covid-19 and post- 3 Covid-19 pneumonia” for consideration in Biomedicines.
We appreciate the reviewers for their insightful comments and we have corrected our original manuscript according to the reviewers’ comments and our point-by-point responses to the reviewer’s comments are included.
Thank you for your consideration.
We look forward to a favorable reply
On behalf of all authors
Sincerely
Claudia Ravaglia
Responds to the reviewer’s comments:
Reviewer 2
Authors provided interesting opinions on the possible role of the IDO1/IDO2 and keynurine pathway in the pathogenesis of Covid-19 and post- 3 Covid-19 pneumonia. Since that is hypothesis, not established fact, I recommend changing of the title in the way implying that (e.g. "Possible role.....") The title has been changed according to the Reviewer’s suggestion
It would be beneficial if authors provide more detailed information on mRNA and protein expression of molecules in question, as well as their distribution regarding healthy tissues.
Details are the object of a specific paragraph (IDO expression in cells and tissues)
It is not clear whether microphotographs constituting Figure 1 are authors' own, obtained in their laboratory. If so, as a part of figure caption authors mut provide details on primary and secondary antibodies (origin, isotype, dilution, manufacturer) and technique used. Also, negative (isotype) controls for each tissue/case should be shown.
The caption has been completed with requested detail and information
Some refinement of wording is needed. I.e. :
Line 244: "....deprivation of Trp mayt (=may) provide...."
Line 246: "...IDO1 expression in normal lung is faint..." , in my opinion, instead of faint expression better choice might be weak expression.
The text has been corrected a general refinement has been performed